# Dating the Common Ancestor from an NCBI Tree of 83688 High-Quality and Full-Length SARS-CoV-2 Genomes

**DOI:** 10.3390/v13091790

**Published:** 2021-09-08

**Authors:** Xuhua Xia

**Affiliations:** 1Department of Biology, University of Ottawa, Marie-Curie Private, Ottawa, ON K1N 9A7, Canada; xxia@uottawa.ca; Tel.: +1-613-562-5718; 2Ottawa Institute of Systems Biology, University of Ottawa, Ottawa, ON K1H 8M5, Canada

**Keywords:** SARS-CoV-2, tip rooting, tip dating, viral evolution, phylogeny, COVID-19, most recent common ancestor

## Abstract

All dating studies involving SARS-CoV-2 are problematic. Previous studies have dated the most recent common ancestor (MRCA) between SARS-CoV-2 and its close relatives from bats and pangolins. However, the evolutionary rate thus derived is expected to differ from the rate estimated from sequence divergence of SARS-CoV-2 lineages. Here, I present dating results for the first time from a large phylogenetic tree with 86,582 high-quality full-length SARS-CoV-2 genomes. The tree contains 83,688 genomes with full specification of collection time. Such a large tree spanning a period of about 1.5 years offers an excellent opportunity for dating the MRCA of the sampled SARS-CoV-2 genomes. The MRCA is dated 16 August 2019, with the evolutionary rate estimated to be 0.05526 mutations/genome/day. The Pearson correlation coefficient (r) between the root-to-tip distance (D) and the collection time (T) is 0.86295. The NCBI tree also includes 10 SARS-CoV-2 genomes isolated from cats, collected over roughly the same time span as human COVID-19 infection. The MRCA from these cat-derived SARS-CoV-2 is dated 30 July 2019, with r = 0.98464. While the dating method is well known, I have included detailed illustrations so that anyone can repeat the analysis and obtain the same dating results. With 16 August 2019 as the date of the MRCA of sampled SARS-CoV-2 genomes, archived samples from respiratory or digestive tracts collected around or before 16 August 2019, or those that are not descendants of the existing SARS-CoV-2 lineages, should be particularly valuable for tracing the origin of SARS-CoV-2.

## 1. Introduction

Two fundamental questions in the study of viral pathogens such as SARS-CoV-2 are the “when” and “where” questions, i.e., when and where specific transmission events occurred from animal to human or from one community to another. One exemplary study addressing both the when and where questions involves the tracing of transmission of HIV-1 from Africa to Caribbean countries and then to North America [1]. Viral phylogeny and dating results from the study are consistent with an early transmission of HIV-1 to Haiti, followed by a subsequent transmission from Haiti to North American populations around the 1960s.

The key parameter in dating the most recent common ancestor is the evolutionary rate. Several studies [2,3,4,5] have used sequence divergence of SARS-CoV-2 and closely related coronaviruses isolated from bats and pangolins to date the most recent common ancestor (MRCA) between SARS-CoV-2 and its most closely related bat or pangolin viruses. These studies dated the MRCA to roughly 50 years ago. For example, the estimated divergence time between SARS-CoV-2 and RaTG13 is 51.71 years [3]. Given a genomic divergence of about 0.04 between Wuhan-Hu-1 and RaTG13, the evolutionary rate is
(1)$$\mu \;\approx \frac{\left(0.04/2\right)\;\times \;30,000}{51.71\times 365}\approx 0.032\;\mathrm{substitutions}/\mathrm{genome}/\mathrm{day}$$

However, this rate is not appropriate for dating the MRCA of sampled SARS-CoV-2 genomes because the estimated evolutionary rate from such studies involves a divergence time of about 50 years. This rate is expected to be smaller than the evolutionary rate estimated from divergence among SARS-CoV-2 genomes [3,4]. The reason is as follows: Sequence divergence between SARS-CoV-2 genomes includes substitution and deleterious mutations that purifying selection has not yet had time to purge off. In contrast, sequence divergence between SARS-CoV-2 and RaTG13 (or another bat- or pangolin-derived virus) is associated with ~50 years of purging by purifying selection, so the evolutionary rate estimated from SARS-CoV-2 genomes only is expected to be greater than those estimated from sequence divergence between SARS-CoV-2 and its relatives from bats or pangolins. It is therefore not appropriate to extrapolate the rate in Equation (1) or any rate estimated from divergence between SARS-CoV-2 and bat- or pangolin-derived CoVs to date the MRCA of SARS-CoV-2 lineages. It is therefore important to estimate the evolutionary rate based on SARS-CoV-2 genomes in the dating of their MRCA.

Dating the MRCA of SARS-CoV-2 lineages based on the genomic divergence of SARS-CoV-2 lineages needs two prerequisites. First, the sample size needs to be very large to overcome the stochastic fluctuation of evolutionary rates among lineages. Second, there needs to be sufficient time for SARS-CoV-2 lineages to accumulate mutations and substitutions. The first prerequisite is well met because of large-scale sequencing efforts worldwide. The second prerequisite is also reasonably well met given the evolution of sampled SARS-CoV-2 lineages over 1.5 years (or longer if the SARS-CoV-2 MRCA originated substantially earlier than December 2019).

A large phylogenetic tree with 86,582 high-quality full-length SARS-CoV-2 genomes was released by NCBI on 3 April 2021. The branch length of the tree is expressed as the number of mutations per genome because one cannot explicitly differentiate between mutation and substitution in the viral genomic data. Therefore, the number of nucleotide differences inferred to have occurred between two genomes is referred to as the number of mutations by NCBI researchers. Such a large tree based on high-quality full-length genomes offers an excellent opportunity for dating the common ancestor of the sampled SARS-CoV-2 genomes. The strength of the data is not just in the number of genomes, but in particular, because the viral lineages have evolved for almost 1.5 years since the Wuhan outbreak. One would not be able to characterize a relationship between sequence divergence and divergence time reliably with only a few weeks or months because few substitution events occur within such a short period. The tree release time proceeded the large-scale administration of COVID-19 vaccines, so the evolutionary rate of the SARS-CoV-2 lineages included in the tree should not be confounded by COVID-19 vaccines as strong selection agents against the virus.

The main difficulty in addressing the “when” question involving SARS-CoV-2 strains is the lack of a close relative to SARS-CoV-2 genomes. There are many nearly identical SARS-CoV-2 genomic sequences with an average sequence divergence of about 0.0002, but the closest relative to SARS-CoV-2, isolated from *Rhinolophus affinis* bat (RaTG13), differs from SARS-CoV-2 sequences by 4% [6]. This gives rise to two problems. First, statistical methods have difficulties detecting which SARS-CoV-2 strain is the closest to RaTG13 because RaTG13 can be placed at multiple locations of the viral tree with little change in the tree likelihood. Second, as I mentioned before, the short branches among SARS-CoV-2 lineages include substitutions and deleterious mutations that purifying selection has not had time to eliminate, whereas the long branch of about 0.04 between SARS-CoV-2 and RaTG13 results from about 50 years of purifying selection along the two lineages.

A commonly used method for tip rooting and ancestor dating (TRAD) has been implemented in software packages including TempEst [7], DAMBE [8], Treetime [9], LSD [10], and Treedater [11]. It is based on well-established least-squares criterion and evolutionary principles [12,13,14,15,16] by using the information of different sampling times of rapidly evolving viruses. While TempEst and LSD are specialised programs, other packages also include many other functions related to phylogenetic reconstruction. One main advantage of this method is its speed. As implemented in TRAD [17], the method can be used to date the MRCA from very large trees such as the NCBI tree with 86,582 leaves in less than an hour with a regular desktop computer (i7-4770 CPG @ 3.4GHz, 32 GB of RAM).

The MRCA of sampled SARS-CoV-2 is dated 16 August 2019, with the evolutionary rate estimated to be 0.05526 nucleotide changes/genome/day. The correlation coefficient between the root-to-tip distance and the collection time is 0.86273. More detailed statistics are presented and discussed later. The phylogenetic tree downloaded from NCBI (https://www.ncbi.nlm.nih.gov/labs/virus/vssi/#/precomptree, accessed on 3 April 2021) is included as tree_21_04_03.nwk within the Appendix A for duplicating the result (because it might be replaced by a still larger tree in the NCBI server). I will first detail the method so that anyone with basic statistical understanding and programming skills can carry out the analysis and obtain the same results without using software written by others. I will then present dating details for SARS-CoV-2 genomes from human and animal samples.

## 2. Materials and Methods

### 2.1. TRAD: Tip Rooting and Ancestor Dating

The method is based on a simple assumption of molecular clock that sequence divergence corrected for multiple hits is on average proportional to divergence time. If the time from the true root is T and the evolutionary distance from the true root is D, then T and D are linearly related. This assumption may not be well met for SARS-CoV-2 because some SARS-CoV-2 lineages appear to evolve faster or slower [2,18] either because of differential selection or mutation. It is not known whether the sampled SARS-CoV-2 genomes might be overrepresented by the fast-evolving or slow-evolving viral lineages. The dating method is based on the faith that evolutionary rate is mainly determined by the rate of neutral mutations and that neutral mutations occur at roughly the same rate across lineages. In addition to this classic molecular clock hypothesis, there are also statistical assumptions that would be obvious given the model presented below.

As illustrated in Figure 1A, if internal Node 1 corresponds to the true root, then its node-to-tip distance (D_1j_, where j = 1, 2,..., 12 correspond to the 12 sequences, labelled S1 to S12 in Figure 1A), according to the molecular clock hypothesis, is expected to be proportional to tip sampling time (T_j_). That is, a longer D_1j_ corresponds to a proportionally later T_j_. Figure 1B shows the 12 T_j_ values and the corresponding D_1j_, D_2j,_ and D_3j_ values for internal Nodes 1, 2, and 3, respectively. If Node 1 is the true root, then the D_1j_ column should be more highly correlated with the T_j_ column (Figure 1B) than other D_ij_ columns. Given that $r_{P}$ between D_1j_ and T_j_ is the largest among all internal nodes, Node 1 is deemed the closest to the true root than other internal nodes. The method also traverses the tree along the branches to find the rooting point with the highest $r_{P}$.

The highest correlation for the tree in Figure 1A is obtained at a point along the branch between Node 1 and Node 5 (Figure 1B), at a distance of 0.0871 from Node 1, with $r_{P}$ = 0.9756. This method allows us to identify approximate rooting position without being bothered by the time to MRCA or the rate of sequence evolution. Once the root is identified, it becomes simple to estimate the evolutionary rate and date the MRCA by linear regression.

Let *T_A_* be the time to MRCA, and μ be the evolutionary rate, the molecular clock gives us the following linear relationship to estimate *T_A_* and μ:(2)$$D=\mu \left(T-T_{A}\right)=-\mu T_{A}+\mu T$$
where *D* (the root-to-tip distance) and *T* (the sampling time of the tips) are the dependent and independent variables in the linear model. $\left(-\mu T_{A}\right)$ is the intercept, and μ is estimated as the slope from the linear regression of *D* on *T*. Consequently, $T_{A}=-Intercept/\mu$. Equation (2) is appropriate for Model I regression, i.e., *T* (sampling date) is known with negligible measurement error, whereas *D* is subject to stochastic fluctuation.

A linear regression of the D_1j_ column on T_j_ column (Figure 1B) yields μ = 0.045 and *T_A_* = 12/9/2019 (9 December 2019). However, the best rooting position (with the highest r) lies along the branch between Node 1 and Node 5, at a distance of 0.087117 away from Node 1 (Figure 1). This root gives μ = 0.046 (95% CI: 0.039 to 0.054) and *T_A_* = 12/10/2019. I will discuss later alternative approaches for estimating variation in *T_A_*.

The accuracy of the dating method illustrated in Figure 1 depends heavily on two factors. The first factor is the sample size. If our sample includes only the red lineages in Figure 1C, then we can estimate only a shallow common ancestor. The dating will be more informative if our sample includes early lineages, e.g., those branching off directly from Node 4 (Figure 1C). The second factor is the time span. If all samples are collected at a single time point, then T is a constant, rendering it impossible to derive a linear relationship between D and T in Equation (2). The NCBI tree released on 3 April covers a time span of almost 1.5 years.

### 2.2. The Phylogenetic Tree of 86582 SARS-CoV-2 Genomes

The SARS-CoV-2 sequences included in the phylogenetic tree meet the following two requirements: (1) sequence length between 29,600 and 31,000 bp after trimming ends which do not align with SARS-CoV-2 reference genome (NC_045512), and (2) the number of ambiguous nucleotides must be <1% in the trimmed sequences. The trimming of the viral sequences at their two ends is necessary because (1) these ending sequences are often missing in SARS-CoV-2 genomic sequences, and (2) they contain more ambiguous codes possibly due to sequencing errors. The infrastructure for viral variation characterization and phylogenetics was created to improve response to emergent viral outbreaks [19]. The high-level application programming interface (API) and low-level phylogenetic functions are implemented in the NCBI C++ toolkit [20].

The leaf labels are in the format of accession.version|host|isolation source|geographic location|collection date|GenBank release date. DAMBE [8] and TRAD [17] use collection dates for dating, so the GenBank release date was removed. I have also abbreviated terms by replacing United_States by US, United_Kingdom by UK, Homo_sapiens by Hs, etc. Out of 86,582 leaves in the tree, 86,431 are from human patients. DAMBE trimmed 2893 leaves that do not have complete dates, e.g., missing month or date specification. I also excluded the SARS-CoV-2 genome (MW737421) from Iran that originally has a collection date of 25 October 2019 but subsequently changed to 11 February 2020. The resulting tree also included as Tree_GoodDates.dnd within the Appendix A contains 83,688 leaves that remain after removing those with incomplete dates. This tree, or any other tree in Newick format with collection date appended to sequence ID, is the only required input for DAMBE [8] and TRAD [17] to perform dating. Other available programs may either not work with a large tree or require additional information beyond the tree, so one may have to use TRAD [17] or write one’s own program to implement the analysis. It takes only four mouse clicks to install TRAD. Click ‘Phylogenetics | TRAD’ and input the Tree_GoodDates.dnd file within the Appendix A to replicate the result in this paper. If you use your own tree and if some of the leaves do not have full date specification, check the ‘Trim OTUs with partial date’ checkbox.

The earliest sample of SARS-CoV-2 was collected in Iran on 12 December 2019. However, because it was submitted to GenBank on 12 April 2021 (ACCN: MW898809), the genome was not included in the NCBI tree released on 3 April 2021. There was a mistaken report of the collection date for a different SARS-CoV-2 sample, originally recorded to have been collected in Iran on 25 October 2019 (ACCN: MW737421). This genome was included in the NCBI tree on 3 April 2021. Upon enquiry on patient information, the submitters of the genome realized that the collection date was incorrectly reported and should be 11 February 2020. They subsequently informed NCBI and updated the collection date on the GenBank record. This suggests the importance of a large tree so that a mistake in data entry will not affect the general conclusion. I excluded this genome (MW737421) from Iran in my dating analysis because of the uncertainty in my mind about its collection date. However, including it has a negligible effect on the estimation results.

## 3. Results and Discussion

### 3.1. Ancestor of Sampled SARS-CoV-2 Is Dated 16 August 2019

Although some SARS-CoV-2 genomes evolve much faster than others, the overall relationship between root-to-tip distance (*D*) and the collection time is strong, with a Pearson correlation coefficient equal to 0.86295 (Figure 2). The mutation rate *μ* = 0.055273/genome/day as estimated by the linear regression equation, and the common ancestor of the sampled SARS-CoV-2 genomes is dated to T_A_ = 16 August 2019 (Figure 2). The resulting rooted tree and the estimated root-to-tip distance for each genome are included as Tree_GoodDates_Rooted.dnd and Tree_GoodDates_RootToTipD.txt, respectively, within the Appendix A. I should emphasise that T_A_ is not for the common ancestor of all SARS-CoV-2 lineages but only for the common ancestor of the sampled SARS-CoV-2 genomes. Only when we have included the most ancient lineages of SARS-CoV-2 would T_A_ approximate the time of origin of SARS-CoV-2.

The dating result of the origin of the MRCA corroborates the finding of two recent publications by Italian scientists [21,22]. The first paper [21] reported detection of SARS-CoV-2 in northern Italy as early as September 2019, but the finding was heavily criticized for using antibody detection instead of the sequence-based detection which is the gold standard in SARS-CoV-2 detection. The second paper [22] took the sequencing approach which is the gold standard in detecting SARS-CoV-2 but reached the same conclusion, with the first positive sample of SARS-CoV-2 collected on 12 September 2019. These findings highlight the importance of analyzing archived samples before the Wuhan outbreak.

Previous estimates of evolutionary rate are based on sequence divergence of SARS-CoV-2 and closely related coronaviruses isolated from bats and pangolins [2,3,4,5]. These studies dated the most recent common ancestor for SARS-CoV-2 and closely related bat or pangolin viruses to about 50 years ago. As some of the deleterious mutations will be purged off over time, the estimated evolutionary rate from such studies involving a divergence time of about 50 years is therefore expected to be smaller than the evolutionary rate estimated from SARS-CoV-2 genomes alone [3,4]. For example, the estimated divergence time between SARS-CoV-2 and RaTG13 is 51.71 years [3], leading to an average evolutionary rate of 0.032 substitutions/genome/day, as shown in Equation (1). It is not appropriate to extrapolate this rate to date the common ancestor of SARS-CoV-2. Evolutionary rates have also been estimated from SARS-CoV, MERS-CoV, or other coronaviruses [2], but again, these rates cannot be extrapolated to SARS-CoV-2.

While the mutation rate *μ* (of 0.055273/genome/day) was taken as a parameter, it may in fact be a variable if a major fraction of mutations is not neutral. For example, effective population size could affect *μ* for selectively advantageous or deleterious mutations. When a virus became highly infectious, its population size increases which results in more efficient selection against slightly deleterious mutations. If a significant proportion of mutations are slightly deleterious, then the increased population size would result in a reduced evolutionary rate. Similarly, large-scale applications of antivirals or vaccines typically result in reduced population size and consequently reduced selection intensity against slightly deleterious mutations. This would result in an increased evolutionary rate. However, the relationship between D and T in Figure 2 appears well described by a linear relationship.

The larger the sample size, the more likely the sample will include early lineages (i.e., lineages that are not descendants of SARS-CoV-2 already represented in the database), and the common ancestor will be dated earlier. If we apply the same dating method to SARS-CoV-2 genomes sampled by 15 January 2020, then T_A_ = 13 December 2019. The genomic data by 21 March 2020 would have T_A_ = 4 December 2019, The genomic data by 8 May 2020 would have T_A_ = 20 October 2019. The large tree from NCBI released on 3 April 2021 pushes T_A_ back to 16 August 2019. As illustrated in Figure 1C, only when our samples include the earliest lineages of SARS-CoV-2 can T_A_ approximate the time of origin of all SARS-CoV-2 viruses. If our sample includes only those red lineages in Figure 1C, then it is impossible for us to date the common ancestor to Node 4.

### 3.2. Assessing the Variation of the Estimated Origin Date of the MRCA

If all SARS-CoV-2 genomes had evolved exactly at μ = 0.055273/genome/day, then *T_i_ − D_i_*/*μ* (where *T_i_* and *D_i_* are the collection time and root-to-tip distance for genome *i*, respectively) would all be equal to 16 August 2019. However, SARS-CoV-2 genomes do not evolve exactly at the same rate, so *T_i_ − D_i_*/*μ* will not be all equal to 16 August 2019. If we designate *T_Ai_* = *T_i_ − D_i_*/*μ,* then variation in *T_Ai_* could serve as an estimate of variation in the estimated ancestral time for the sampled SARS-CoV-2 genomes (Figure 3), which includes mean, standard deviation, and 95% confidence limits). This visual display is perhaps more appropriate than using the standard error of regression coefficients to attach confidence intervals to the origin of the MRCA. As the very large sample size, the confidence interval for the date of the MRCA origin would be within two days which could be highly misleading because a large number of genomes are not independent samples from the same distribution. Figure 3 also highlights the point that evolutionary rate, although often estimated as a parameter, it is associated with much sampling variation.

There is no statistically sound way of arriving at degrees of freedom with coancestry in the form of a phylogenetic tree. If we had a star tree, then all lineages evolve independently so that n lineages would have *n* independent points. There are only about 50 early SARS-CoV-2 lineages which appear to radiate out similar to a star tree. All the other SARS-CoV-2 lineages appear to descend from these early lineages. Therefore, a conservative estimate of the standard error would be $SD/\sqrt{n}=73/\sqrt{50}=10.324$ instead of the potentially misleading $73/\sqrt{83668}$. This conservative estimate of the standard error of 10.324 yields a 95% confidence interval of T_A_ being 16 August 2019 ± 20 days. Dating results from this large tree provide concrete and convincing support to the hypothesis that SARS-CoV-2 might have been transmitted cryptically among human populations months before the viral outbreak [23,24].

Another complication in attaching confidence intervals to the dated origin of sampled SARS-CoV-2 genomes is that the distribution in Figure 3 may not be a single distribution. The distribution in Figure 3 could be a mixture of two or more distributions. For example, if an overwhelming majority of SARS-CoV-2 genomes were descended from a common ancestor on 16 August 2019, but a few SARS-CoV-2 genomes branches off from an earlier ancestor, e.g., 30 March 2019, the effect of these few genomes on the estimation of overall evolutionary rate and the ancestral date would be negligible, especially if such genomes contain sequencing errors that prevent their sequence divergence from being accurately calculated. However, if there are a sufficient number of such genomes, they would give rise to an irregular bump in the distribution based on individual genomes (Figure 3). The red arrow in Figure 3 indicates this possibility of a mixed distribution. In this context of a mixture distribution, we could only state that an overwhelming majority of the sampled SARS-CoV-2 genomes were descended from a common ancestor dated 16 August 2019. It does not exclude the possibility that some of the SARS-CoV-2 genomes diverged earlier.

If we could extrapolate my estimated μ of 0.055273/genome/day to the evolution of SARS-CoV-2 and its closest relative RaTG13 [6], then the common ancestor of SARS-CoV-2 and RaTG13 can be dated to (D_SARS-CoV-2,RaTG13_/2)/μ. D_SARS-CoV-2,RaTG13_, the average distance between SARS-CoV-2 and RaTG13, is ~0.04/site which is translated to ~1200/genome. Therefore,


(3)
$$\frac{\frac{D_{SARS-CoV-2,RaTG13}}{2}}{\mu }\;=\frac{\left(\frac{1200}{2}\right)}{0.055273}\;=\;29.7\;\mathrm{years}$$


Note that this is an underestimate because the number of nucleotide differences between SARS-CoV-2 genomes includes both mutations and substitutions, whereas the number of nucleotide differences between SARS-CoV-2 and RaTG13 should include mainly substitutions. The rate of substitutions obviously is lower than the rate of both mutations and substitutions, which implies a smaller μ than 0.0055273. In other words, SARS-CoV-2 and RaTG13 should have diverged substantially more than 29.7 years. More extensive dating of the ancestor of SARS-CoV-2 and other SARS-related coronaviruses has been carried out previously [3,4], with an estimated divergence time earlier than 29.7 years. However, it is a long gap in SARS-CoV-2 evolution even with 29.7 years in which we have no phylogenetic information. What is needed to fill in the gap are viral genomes that are not descendants from existing SARS-CoV-2 lineages but more closely related to SARS-CoV-2 than the bat RaTG13 genome.

I should emphasise again that it is inappropriate to extrapolate the evolutionary rate obtained from SARS-CoV-2 genomes to date the divergence between SARS-CoV-2 and its relatives isolated from bats and pangolins, nor is it appropriate to extrapolate the evolutionary rate estimated from sequence divergence between SARS-CoV-2 and bat/pangolin-derived relatives to the evolutionary rate of SARS-CoV-2 genomes. To date the MRCA of SARS-CoV-2, one needs to use SARS-CoV-2 genomes, a strategy also used in this paper.

### 3.3. Outliers in Figure 2

There are a number of outliers in Figure 2 that deserve discussion, as outliers often represent a different way of conferring the truth. I previously mentioned the stochastic effect on evolutionary rate due to differential accumulation of mutations in different viral genomes. However, this does not satisfactorily explain the outliers in Figure 2. There are a number of genomes that have evolved too fast from the expectation. This is also reflected in the small distribution bump in Figure 3 (pointed to by the red arrow). This brings up the second source of outliers. That is, those outliers may either evolve extremely fast or represent descendent lineages from an even earlier common ancestor.

I will illustrate this with an outlier representing the SARS-CoV-2 genome (MW795884, circled in red in Figure 2). This genome belongs to the D614G strain that was recorded to increase rapidly in frequency in Europe and then in North America since March 2020, displacing the original Wuhan D614 strain in multiple locations [18]. As the D614G strain was first noted in Europe, people have often implicitly assumed that it was transmitted to North American from Europe. However, this outlier in the red circle in Figure 2 was from a sample collected in Utah on 13 January 2020, suggesting that it might have occurred in the US before Europe. The genomic sequence was submitted by Utah Public Health Laboratory. It already has all four characteristic mutations of the D614G strain described by Korber et al. [18]. There are a total of 25 mutations between this Utah genome and the reference genome from Wuhan but no insertions or deletions. This strongly suggests that the D614 and G614 were co-circulating at the time of the Wuhan viral outbreak, otherwise, one would have to assume the accumulation of 25 mutations over 18 days (a mutation rate higher than the highest in any RNA viruses). The first COVID-19 patient in Utah was previously recorded on 28 February [25], involving a patient who was a passenger on the cruise ship *Diamond Princess* [26]. The SARS-CoV-2 sequence from a sample collected on 13 January 2020 suggested much earlier COVID-19 occurrence in Utah.

I sent an enquiry through the Utah Public Health Laboratory web portal (https://uphl.utah.gov/contact/, accessed on 1 September 2021) about sampling details of the SARS-CoV-2 record (MW795884). Surprisingly, the collection date was simply deleted from the GenBank on 4 September 2021. I have asked about sampling information for only three Genbank records (two others being MW737421 and MW898809, with original collection dates on 25 October 2019 and 12 December 2019, respectively). In all three cases, the collection date was either deleted or revised to a later date upon my inquiry. This highlights another possible cause for the outliers, i.e., they represent GenBank records with potentially wrong information. It also suggests an need for NCBI to document the revision history of sequence records and to discourage researchers who submit sequence data to NCBI without due care.

### 3.4. Dating the Origin of MRCA Based on SARS-CoV-2 from Cats

The NCBI tree also includes 150 SARS-CoV-2 genomes not from human patients. These include samples from cats, minks, tigers, lions, dogs, and hamsters, as well as environmental samples such as sewage, door handle, air, etc. Cats constitute the only species with sample collection time spanning almost the entire period of human infection (Figure 4). The regression of D over T leads to an estimate of μ = 0.053 mutations/genome/day and T_A_ = 30 July 2019 (Figure 4). This exemplifies the point that a small number of data points covering a long time is more valuable than a large number of data points from a single time point. One cannot use the TRAD approach to properly date the MRCA when there are only one or two collection time points, even if each time point is represented by many genomes, as is the case for farmed minks in US and Europe.

The correlation between D and T (r = 0.98464, Figure 4) for cat-derived SARS-CoV-2 is higher than that in Figure 2 (r = 0.8630). If cats are infected sporadically through humans so that cat-derived SARS-CoV-2 genomes are simply a random sample of the human SARS-CoV-2 over time, then we expect the correlation between D and T to be the same for human-derived and cat-derived SARS-CoV-2 genomes. If we bootstrap 10 pairs of D and T from human-derived SARS-CoV-2 genomes 1000 times, the number of times a correlation between D and T is equal or higher than 0.98464 is about 6, i.e., p = 6/1000 = 0.006. However, this could be explained as follows: Suppose that SARS-CoV-2 evolution follows a strict molecular clock so that most of the variation in D for a given T is due to sequencing error. If sequencing error is on average smaller for cat-derived genomes than human-derived genomes, then we obtain a higher correlation for cat-derived data than for human-derived data. The cat-derived SARS-CoV-2 genomes were indeed sequenced by leading viral laboratories (School of Public Health of the University of Hong Kong, Department of Veterinary Pathology of the University of Liege, Virology of Ecole Nationale Veterinaire de Toulouse, Diagnostic Virology Laboratory, USDA, Department of Biosystems Science and Engineering, ETH Zurich, and Bureau of Public Health Laboratories in Florida, USA). In contrast, some human-derived SARS-CoV-2 genomes were sequenced by laboratories in developing countries with multiple ambiguous codes. Although NCBI researchers have already excluded many of these sequences in choosing full-length high-quality genomes to build the tree, some human-derived SARS-CoV-2 genomic sequences are just barely above the quality threshold used in choosing genomes to build the phylogenetic tree.

The first recorded SARS-CoV-2 outbreak in minks occurred in the Netherlands in a mink farm designated NB1, in late April of 2020 [27]. If there were a powerful mink government that insists that investigations must be conducted only in this NB1 farm to find the origin of SARS-CoV-2, we would consider this mink government foolish. We would be eager to advise this mink government that the origin of SARS-CoV-2 was likely beyond the NB1 farm and that it would make better progress by surveying those other mammalian populations, including that of a particular species named *Homo sapiens* that are just one breath away from them, rather than by focusing on the NB1 farm. For the same reason, it is not reasonable to insist that the origin of SARS-CoV-2 could be found at the site of the first viral outbreak in Wuhan, given the evidence that the common ancestor of the sampled SARS-CoV-2 genomes originated as early as mid-August 2019. We could make much better progress by increasing surveillance of wildlife populations that could be just a breath away from us than by focusing only on Wuhan. Many biologists have called for an increased effort of monitoring wildlife populations [23,28,29,30]. The significance of such surveillance surpasses our understanding of the origin of SARS-CoV-2. For example, coronaviruses isolated from wildlife such as pangolin were later found to have a high potential to bind human ACE2 receptors, gain cell entry, and infect humans [31]. Therefore, surveillance provides an efficient way of identifying emergent viral pathogens and preventing future viral outbreaks.

### 3.5. The Identification of Root Is Harder Than Dating the Common Ancestor

Identification of the root on the viral phylogenetic tree is more equivocal than dating the common ancestor. If we throw a coin 1000 times and have 500 heads, the proportion of observing heads is 0.5, and the data are the most consistent with the hypothesis that the probability of having heads (P_head_) is 0.5 because this P gives us the highest likelihood. However, we cannot say that the hypothesis of P_head_ = 0.5 is significantly better than the alternative hypothesis of P_head_ = 0.4999 or P_head_ = 0.5001. In the same way, we identify the root by the Pearson correlation (r) between root-to-tip distance (D) and collection time (T). The candidate rooting location that gives us the highest r is chosen as the root. The root of the tree (Figure 5), indicated by a red dot, has the highest r = 0.86295. However, its neighboring nodes have r almost just as large. Some of them have r values differing from the maximum r only after the fourth digits after the decimal (Figure 5B). We, therefore, have the same problem as discriminating between one hypothesis with P_head_ = 0.5 and alternative hypotheses with P_head_ = 0.4999 and P_head_ = 0.5001. Rather than identifying a specific rooting point, one can only identify a general “rooting region” on the tree (somewhat equivalent to the confidence interval of a point estimate), even with the very large tree of 83688 genomes.

Two of the early SARS-CoV-2 genomes isolated from Wuhan patients are close to the root (Figure 5A), and other viral strains close to the root but isolated elsewhere can be traced to Wuhan as well. That the root is within the large Asian clade is not surprising based on the existing data and is unlikely to change given existing data, as long as new SARS-CoV-2 genomic sequences are mostly descendants of the existing viral lineages.

An alternative is to root the tree with the bat/pangolin-derived viral sequences [3,4,32,33,34]. All these bat-/pangolin-derived viral genomes share D614, so D614G likely is derived, although the D614G change may have occurred earlier than previously thought. The root of SARS-CoV-2 was consistently placed within the large Asian clade if we use bat-/pangolin-derived viral genomes to root the tree.

The problem of rooting with bat-/pangolin-derived viral genomes is that the seemingly giant tree of SARS-CoV-2 essentially shrinks into a dot when the bat-/pangolin-derived viral sequences are added. The reason is that SARS-CoV-2 sequences are, on average, 99.9% identical, which is almost negligible compared to sequence divergence of 96% between SARS-CoV-2 and the closest RaTG13 from bat. The consequence is that one can put the root in many different locations of the SARS-CoV-2 tree with the resulting tree lnL being essentially the same. Another associated problem is that the evolutionary parameter estimates depend heavily on the long branch leading from RaTG13 (or other bat-/pangolin-derived sequences) to SARS-CoV-2 because most of the substitutions occurred along this branch. This suggests that one is extrapolating the evolutionary rate along this branch to the evolutionary rate among SARS-CoV-2 lineages.

I discussed the weakness of this approach of rooting with bat-/pangolin-derived viral genomes in the introduction of the paper, which motivated my approach of estimating the evolutionary rate and dating based only on SARS-CoV-2 sequences. My study corroborates those previous studies by showing that the root is within the large Asian clade without using bat-/pangolin-derived viral genomes to root the SARS-CoV-2 tree.

I should mention that viral genomic divergence is typically gradual, so one should expect some viral genomes more closely related to SARS-CoV-2 than RaTG13. WHO has called upon scientists to sequence archived samples, to test the presence of SARS-CoV-2-specific antibodies in blood samples, and to increase animal surveillance [35]. The approach of going back to archived samples has been shown to be fruitful. For example, the detection of ZIKV antibodies in archived samples discovered the presence of ZIKV positive cases in the United States much earlier than commonly thought [36]. Two recent studies [21,22] based on archived samples identified SARS-CoV-2 in samples collected as early as September 2021 in northern Italy. Although the first study [21] used the antibody approach which may have false positives, the second paper [22] used the sequencing approach which is the gold standard for SARS-CoV-2 detection. Greater accuracy in dating the MRCA of SARS-CoV-2 than achievable through existing data is to sample viral genomes that are more closely related to SARS-CoV-2 than RaTG13 but are not descendants of existing SARS-CoV-2 lineages. If, after exhaustive search, RaTG13 remains the closest relative to SARS-CoV-2 without any evolutionary intermediate, then the hypothesis of a lab leak could gain steam. However, as being pointed out before [37], nature is not obliged to keep intact all evolutionary intermediates for us to discover.

Addressing the question of when and where SARS-CoV-2 originated is difficult partially because SARS-CoV-2 has already spread widely before the lockdown of Wuhan on 23 January 2020 [38]. This has the undesirable consequence that an overwhelming majority of SARS-CoV-2 genomes are from descendants of those that caused the initial outbreak in Wuhan, rendering it more difficult to trace the evolutionary history back to earlier SARS-CoV-2 lineages. The problem is illustrated in Figure 1C. If SARS-CoV-2 samples are limited to those red lineages (Figure 1C), then we can only date a shallow common ancestor. Viral samples beyond those Wuhan lineages and their descendent lineages are badly needed to increase the resolution and to identify ancestral cladogenic events earlier than the estimated T_A_ in this study. Future sampling of SARS-CoV-2 should aim to sample beyond those in Figure 5. Given that the two closest relatives of SARS-CoV-2 were isolated from bats in Yunnan, China, with RaTG13 [6] from Mojiang Hani County and RmYN02 [39] from Mengla County, one should expect bats in these regions most likely to harbor viruses representing early lineages of SARS-CoV-2. In particular, the Tongguan mineshaft in Mojiang, Yunnan, where miners suffered from pneumonia-like diseases similar to COVID-19, should be a focal point of investigation [40].

The number of COVID-19 infection cases has decreased substantially after April 2021, especially after the mass vaccination program, although there is a recent but mild increase in reported cases of infection. The decrease in the viral population size is likely to increase the evolutionary rate of the viral lineages. For example, if most mutations are slightly deleterious, then the reduction in population size will increase the chance of fixation of these slightly deleterious mutations and consequently increase the evolutionary rate [41,42]. SARS-CoV-2 exhibits strong codon usage bias [43], so mutations from a major codon to a minor codon would qualify as being slightly deleterious. Regardless of what factors contribute to the increase in evolutionary rate, such increases are likely to bias the estimate of the MRCA based on the linear model in Equation (2). As illustrated in Figure 6, a linear relationship between D and T, reflecting a molecular clock, would date the common ancestor to 10 July 2019, but a recent increase in evolutionary rate would bias the dating of the common ancestor to 15 December 2019. A model accommodating the change of evolutionary rate over time may be needed if SARS-CoV-2 genomes sampled after April 2021 are included in dating. However, the relationship between Root-to-tip distance and collection time, as shown in Figure 2 and Figure 4, is approximately linear.

## 4. Conclusions

In summary, I dated the common ancestor of sampled SARS-CoV-2 genomes to 16 August 2019 with a large tree of 83688 genomes. Given the ~1.5-year time span since the Wuhan outbreak, and that the viral evolutionary rate not affected by the large-scale vaccination after April 2021, the date estimate should be reasonably accurate. However, to achieve a more accurate estimation of SARS-CoV-2 origin, one needs to sample viral genomes that are more closely related to SARS-CoV-2 than RaTG13 but are not descendants of existing SARS-CoV-2 lineages.

## Figures and Tables

**Figure 1 viruses-13-01790-f001:**
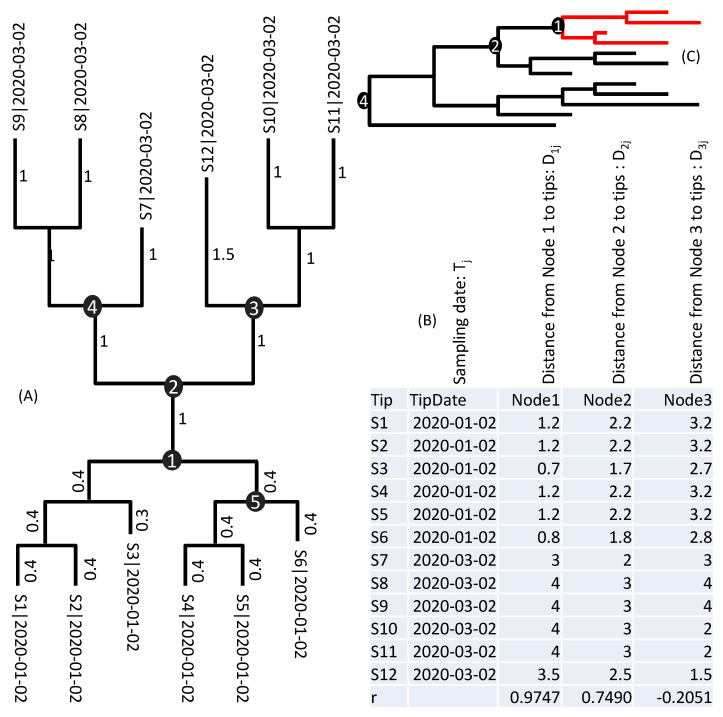
Rationale for identifying which internal node is closest to the true root: (**A**) a tree with 12 tips (S1 to S12) whose sampling dates are appended to sequence names. Branch lengths are shown next to branches, and five internal nodes numbered; (**B**) node-to-tip distances and sampling date between internal Nodes 1, 2, and 3 to the 12 tips. The best internal node is Node 1 with the highest r, but the best rooting position is along the branch between Nodes 1 and 5, at a distance of 0.087117 from Node 1; (**C**) samples including descendants of Node1 can only date their MRCA at Node 1. Only when the most ancient lineage is sampled can we date the MRCA at node 4.

**Figure 2 viruses-13-01790-f002:**
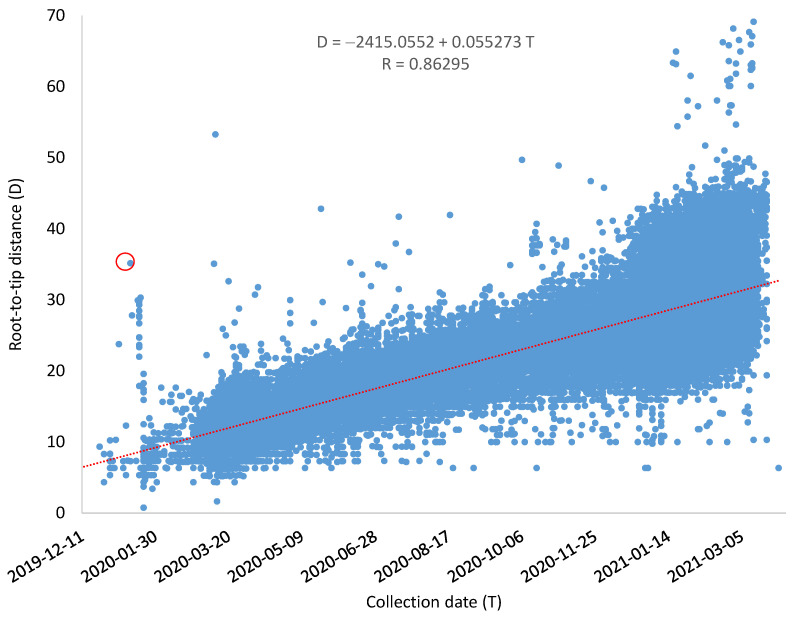
Regression of root-to-tip distance (D) on sampling date (T) of 83688 SARS-CoV-2 genomes, for dating the origin time of the most recent common ancestor (T_A_) of the sampled SARS-CoV-2 genomes and estimating the rate (μ) of sequence evolution from the regression equation (*D* = *a + bT*). *μ* = *b* = 0.055273 mutations/genome/day, and *T_A_* = *−a/b* = 43,693.3 = 15 August 2019 (where time is counted from 1 January 1900 as day 1, 2 January 1900 as day 2, etc.

**Figure 3 viruses-13-01790-f003:**
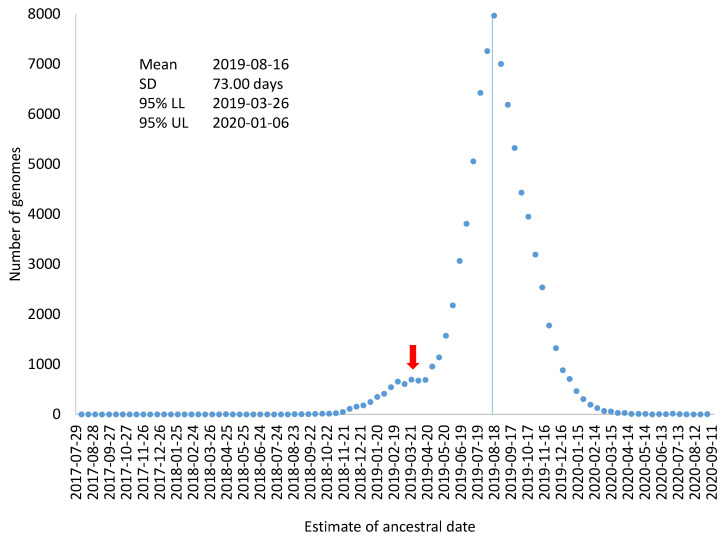
Frequency distribution of T_Ai_ (=T_i_ − D_i_/μ) for characterizing variation in T_A_ (the time for the origin of the common ancestor of the sampled SARS-CoV-2 genomes). The mean, standard deviation (SD), and 95% confidence interval are shown. The red arrow indicates the possible existence of another distribution, i.e., the distribution might be a mixture of two distributions.

**Figure 4 viruses-13-01790-f004:**
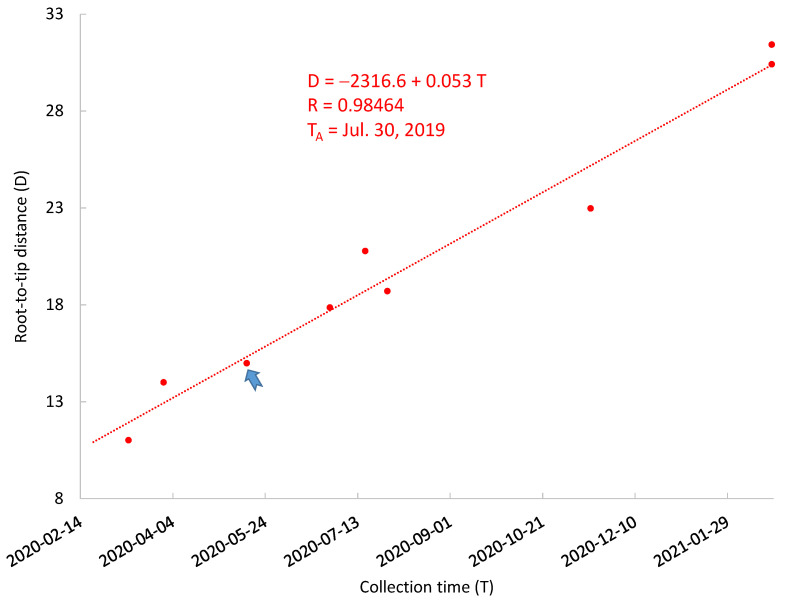
The relationship between root-to-tip distance (D) and SARS-CoV-2 collection time (T) for non-human samples. A regression line is fitted to 10 samples from 10 infected cats (red dots). The blue arrowhead points to two overlapping red dots. The ancestor of cat SARS-CoV-2 was dated 30 July 2019.

**Figure 5 viruses-13-01790-f005:**
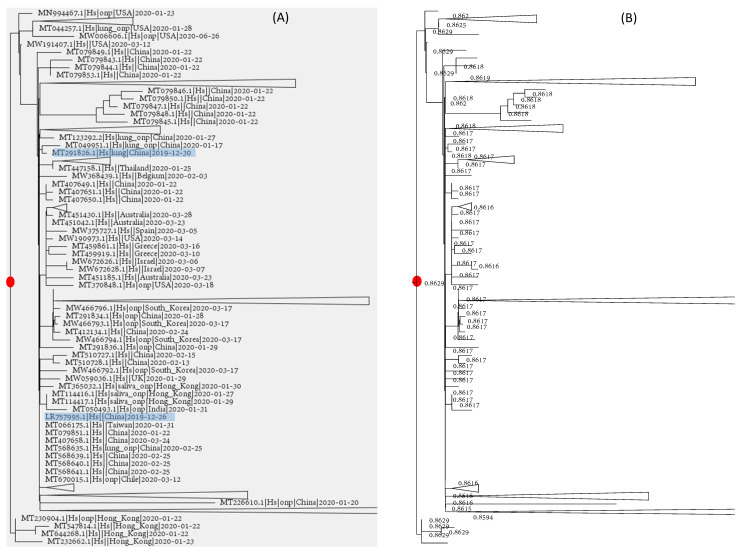
Root identification is hard: (**A**) the tree of 83688 SARS-CoV-2 genomes, with the root indicated by a red dot and many clades collapsed to ease visualisation of viral genomes close to the root; (**B**) the same tree with different candidate rooting positions with Pearson correlation coefficient (r) between the root-to-tip distance and collection time. The red dot indicates the rooting point with the highest r, but a number of alternative root positions have nearly identical r.

**Figure 6 viruses-13-01790-f006:**
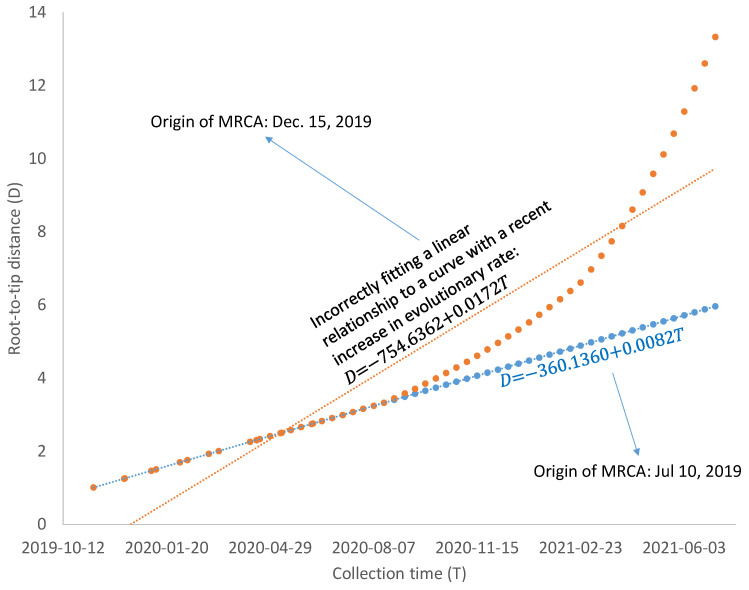
Illustration of bias in dating the most recent common ancestor (MRCA) of SARS-CoV-2 lineages based on Root-to-tip distance (D) and collection time (T), when the evolutionary rate has increased recently.

## Data Availability

Not applicable.

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
