# Peer review of "Dating the Common Ancestor from an NCBI Tree of 83688 High-Quality and Full-Length SARS-CoV-2 Genomes"

_viruses, 2021, doi:10.3390/v13091790_

Round 1

Reviewer 1 Report

Comments on  "Dating the common ancestor from an NCBI tree of 83688
high-quality and full-length SARS-CoV-2 genomes" by Xuhua Xia.

This paper attempts to date the origin of SARS-CoV-2 in humans and
estimates this date to be Aug 16 2019.

The paper nicely gives all of the data that is necessary to
completely reproduce the study.  More papers should be easily
reproducible like this one.

The paper's methods are simple and straightforward but I fear that
the problem is not as simple as the approach used here and requires
a more nuanced approach.  The major problem that I see with the
manuscript is the SARS-CoV-2 is a virus that has just switched host
and is now evolving in humans.  This suggests that it is under
pressure to to adapt and to change to the characteristics of the
host.  This includes not only selected changes but also neutral
changes as the virus uses some of the host machinery to replicate.
Thus I would expect the rates of evolution to be variable and either
increasing or even possibly decreasing.  But there is little
evidence for or against this in this manuscript. 

A second major problem is that the genomes are highly related and
their phylogenetic relationships should be accounted for but again
there is little attempt to do so in this manuscript.

There are obvious outliers at the beginning and end of the graph in
Figure 2.  Perhaps these should be discussed.

Pg 7.  It is not sufficient to simply adjust n rather than to
correct for the obvious correlation between samples.  In addition,
adjusting n to account for the very few samples that were poorly
placed in the tree is not a correct approach.

Bottom Pg 8, top Pg 9.  Human derived sequences are well sequenced.
You state that cat derived sequences are more accurately sequenced.
But you don't provide any evidence for this at all.  There are
programs designed to estimate sequencing errors.  Yet you have not
applied these.

Pg 9.  Paragraph beginning 'American minks'.  You state that a
regression is not suitable.  And then you go ahead and do it.

There is no discussion of the obvious differences between American
and European minks in Fig 4.  This suggests that other processes
might be entering the problem.

Regarding Fig 6.  I do anticipate that a non-linear molecular clock
would be most appropriate or at least should be rigorously refuted.

Minor changes:
Pg 2. Paragraph beginning 'A large phylogenetic'.  
Change 'NCBI researchers).' to 'NCBI researchers.'
Change 'months because of few substitution' to 'months because few substitution'

Pg 3. Paragraph beginning 'The MRCA of'.
'because it might be replaced by a still larger tree' -> The justification
should perhaps be for repeatability rather than that a better tree will
be found.

Author Response

I thank the reviewer for the insightful comments which bring several issues into focus and results in extensive rearrangement of the "Results and Discussion" section. My responses are in bold italic.

This paper attempts to date the origin of SARS-CoV-2 in humans and estimates this date to be Aug 16 2019. The paper nicely gives all of the data that is necessary to completely reproduce the study.  More papers should be easily reproducible like this one.

 The paper's methods are simple and straightforward but I fear that the problem is not as simple as the approach used here and requires a more nuanced approach.  The major problem that I see with the manuscript is the SARS-CoV-2 is a virus that has just switched host and is now evolving in humans.  This suggests that it is under pressure to to adapt and to change to the characteristics of the host.  This includes not only selected changes but also neutral changes as the virus uses some of the host machinery to replicate. Thus I would expect the rates of evolution to be variable and either increasing or even possibly decreasing.  But there is little evidence for or against this in this manuscript.

There is indeed several factors can alter the rate of evolution. For example, when a virus became highly infectious, its population size increases which results more efficient selection against slightly deleterious mutations. If a significant proportion of mutations are in fact slightly deleterious, then the increased population size would result in a reduced evolutionary rate. Similarly, large-scale applications of antivirals or vaccines typically result in reduced population size and consequently reduced selection intensity against slightly deleterious mutations. This would result in an increased evolutionary rate.

However, for the SARS-CoV-2 data, the relationship between D and T is approximately linear, as shown in Fig. 2 (and Fig. 4 for cats). This could be explained by the period during which the samples were taken. The viral population size was already quite large almost right after its outbreak, and it remained large by Apr. 3 when the viral tree was released because no effective antivirals or vaccines were available to dramatically reduce viral population size. It is also possible that most mutations are selectively neutral, so the selection scenario above does not apply. I have expanded the discussion on factors affecting evolutionary rate.

The simplest approach to test whether the rate is monotonically increasing or decreasing over time is to use a quadratic function between D and T. The rest of the algorithm can remain the same. One traverses along the nodes and branches, considering every point as a possible root, and finds the root at which the quadratic function best explains the relationship between D and T. The non-linear component began to increase from May 20, 2021, but not before. The tree I used has sequences by Apr. 3, 2021 and does not show any increase or decrease in evolutionary rate, which is not surprising given Fig. 2 and Fig. 4 (for cats).

A second major problem is that the genomes are highly related and their phylogenetic relationships should be accounted for but again there is little attempt to do so in this manuscript.

The dating is entirely based on phylogenetic relationships and branch lengths as I have explained in Fig. 1. The algorithm traverses the tree nodes and branches, considering every point along all branches as a possible root. The point from which the root-to-tip distance (D) that best explains the variation in collection date (T) is the estimated root. This is why the algorithm is fairly computationally intensive. It is also partly for this reason that my TRAD and DAMBE are the only ones that can handle such a large tree (but TRAD is much faster than DAMBE).

There are obvious outliers at the beginning and end of the graph in Figure 2.  Perhaps these should be discussed.

This is an excellent point as outliers often represent a different way of conferring the truth. I mentioned two possible sources of outliers. The first is simply stochastic variation in evolutionary rate. Some viral genomes can accumulate several mutations over weeks, but some others do not change for months. For example, one strain (Wuhan|WIV06|EI402129|2019-12-30), sampled on Dec. 30, 2019, was identical to 40 other genomic sequences, isolated from patients in Thailand on Jan. 8 and 13, in Japan on Jan. 23, South Korea on Feb. 1, Canada on Mar. 3 and Ireland on Mar. 4. These genomic sequences differ only in the presence/absence of the two extremes in the sequenced genomes. Excluding the short sequence at the two ends that are not shared among sequenced genomes, these genomic sequences are identical. When genomes do not change but are sampled over different times, the root-to-tip the distance (D) does not change but the collection date (T) differ. In a plot of D versus T, this would generate a horizontal row of dots with the same D but different T. This is why I emphasized the importance of a large number of viral lineages to average out the stochastic effects on viral evolution. However, this does not satisfactorily explain the outliers in Fig. 2. There are a number of genomes that have evolved too fast (diverged too much from the putative root) from the expectation based on the majority of sequences. This is also reflected in the small distribution bump in Fig. 3. This brings up the second source of outliers. That is, those outliers may either evolve extremely fast, or represent descendent lineages from an even earlier common ancestor. I discussed this briefly with respect to Fig. 3.

I might illustrate this second source of outliers with the SARS-CoV-2 genome (GenBank accession MW795884) circled in red in Fig. 2. This genome belongs to the D614G strain that has increased rapidly in frequency since March 2020 and displaced the original D614 strain that caused the outbreak in Wuhan. Because the D614G strain was first noted in Europe, people have often implicitly assumed that it came to North American from Europe (partly because Trump left the airport open to airplanes from Europe after closing air travel from China). However, this outlier in red circle in Fig. 2 was from a sample collected in Utah on Jan. 13, 2020, suggesting that it might have occurred in US before Europe. The genome already has all four characteristic mutations of the D614G strain described by Korber et al. {Korber, 2020 #57990}. There are a total of 25 mutations between this Utah genome and the reference genome from Wuhan, but no insertions of deletions. This strongly suggests that the D614 and G614 were co-circulating at the time of Wuhan viral outbreak, otherwise one would have to assume the unrealistic accumulation of 25 mutations over 18 days (a mutation rate higher than the highest in any RNA viruses).

In summary, the outlier is likely misplaced in the phylogenetic tree. Given a total of 83688 genomes on the tree, it is highly likely that some were misplaced on the tree and contribute to the outliers. Interestingly, those viral genomes that are sequenced in US but can be traced to visitors to China are not outliers. I have added more to the discussion.

Pg 7.  It is not sufficient to simply adjust n rather than to correct for the obvious correlation between samples.  In addition, adjusting n to account for the very few samples that were poorly placed in the tree is not a correct approach.

The different values of n here is to illustrate different statistical views on the reduction of degree of freedom with non-independent data. If the tree is a star tree, then the (D, T) for each lineage is independent of each other, excluding viral recombination. I have rephrased the paragraph.

Bottom Pg 8, top Pg 9.  Human derived sequences are well sequenced. You state that cat derived sequences are more accurately sequenced. But you don't provide any evidence for this at all.  There are programs designed to estimate sequencing errors.  Yet you have not applied these.

I should have been more explicit. The 10 cat-derived SARS-CoV-2 genomes were sequenced by the following laboratories, which are all leading viral research centers in the world.

MT628700.1

School of Public Health, The University of Hong Kong

MT747438.1

Department of Veterinary Pathology, University of Liege

MT709104.1

Virology, Ecole Nationale Veterinaire de Toulouse, 23 chemin des capelles, Toulouse 31076, France

MT709105.1

Virology, Ecole Nationale Veterinaire de Toulouse, 23 chemin des capelles, Toulouse 31076, France

MW263334.1

Diagnostic Virology Laboratory, USDA.

MW263335.1

Diagnostic Virology Laboratory, USDA.

MW263337.1

Diagnostic Virology Laboratory, USDA.

MW600500.1

Department of Biosystems Science and Engineering, ETH Zurich

MW725609.1

Bureau of Public Health Laboratories, Florida, USA

MW725611.1

Bureau of Public Health Laboratories, Florida, USA

Among the 10 genomic sequences from cat-derived SARS-CoV-2, the first seven genomes have no ambiguous codes. The last three have only a few unresolved bases. In contrast, some human-derived SARS-CoV-2 genomes were sequenced by laboratories in developing countries. Although NCBI researchers have already excluded many of them in choosing full-length high-quality genomes to build the tree, some human-derived SARS-CoV-2 genomic sequences are just barely above the quality threshold used in choosing genomes to build the phylogenetic tree.

Pg 9.  Paragraph beginning 'American minks'.  You state that a regression is not suitable.  And then you go ahead and do it.

I should be more explicit. It is OK to characterize a line with two clusters of points, but it would be wrong to estimate variability of the estimates assuming the cluster of points as independent.

There is no discussion of the obvious differences between American and European minks in Fig 4.  This suggests that other processes might be entering the problem.

This is a good point. SARS-CoV-2 from American minks all belong to the G614 type in both clusters. In contrast, the early cluster of SARS-CoV-2 from European minks is made of a mixture of D614 and G614 types, but the late cluster of SARS-CoV-2 from European minks all belongs to G614 type. The common ancestors of the European and American minks are dated to Dec. 15, 2019 and Dec. 09, 2019, respectively, consistent with interpretation that their respective common ancestors are descendants of the common ancestor of all sampled SARS-CoV-2 genomes.

Regarding Fig 6.  I do anticipate that a non-linear molecular clock would be most appropriate or at least should be rigorously refuted.

I used a descriptive quadratic function as an alternative. For trees after May 20, the non-linear component gradually becomes more significant, but not before. Fig. 2 and Fig. 4 (for cats) show no indication of non-linear molecular clock. I have emphasized this point in the manuscript.

Minor changes:

Pg 2. Paragraph beginning 'A large phylogenetic'. 

Change 'NCBI researchers).' to 'NCBI researchers.'

Corrected.

Change 'months because of few substitution' to 'months because few substitution'

Corrected

Pg 3. Paragraph beginning 'The MRCA of'.

'because it might be replaced by a still larger tree' -> The justification should perhaps be for repeatability rather than that a better tree will be found.

Yes, rephrased.

Reviewer 2 Report

The manuscript “Dating the common ancestor from an NCBI tree of 83688 high-quality and full-length SARS-CoV-2 genomes” details the estimation the the ancestor of sampled SARS-CoV-2 using a large tree available in NCBI.

While the paper technically is sound, I worry about its novelty (tip-dating has been performed in other publications) and on an opposite view, I also worry about its major novelty: the early date obtained for the age of the ancestor of SARS-CoV-2.

The major advantage reported here is the fact that the author used a tree with data collected through 1.5 years. I wonder if this is an advantage. The circulating viruses 1.5 years after the first reported sequences is quite different from the early one in late 2019 and early 2020. As the author mentions “... some SARS-CoV-2 lineages appear to evolve faster or slower [4,19] either because of differential selection or mutation.” The history of the pandemic for 1.5 years shows a bias that could affect mutation rates estimates: often lineages that have a greater number of mutations are the ones that would have an increased spread with increased transmissibility against non-mutated lineages similar to the ancestors. I find more believable age estimates based on a small tree with only early sampled sequences (more similar to the SARS-CoV-2 ancestor lineage against a large tree with a great variability of lineages as reported here.

The author even performed some analyses on this using only sampled genomes up to Jan. 15 2020, Mar. 21 2020, May 8 2020 and Apr. 3 2021 leading to consecutively earlier TMRCA with dates of  Dec. 13 2019, Dec. 4 2019, Oct. 20 2019 and Aug. 16 2019. This is an interesting trend. The author claims that “The larger the sample size, the more likely the sample will include early lineages, and the common ancestor will be dated earlier.” but that statement makes little sense as, from my knowledge, later trees do not earlier lineages, only descendants of the same early lineages detected up until early 2020.

The author bases the reliability of the early age on a recent report from Italy claiming the presence of SARS-CoV-2 lineages since September 2019 in Northern Italy. I do not think that report has been properly peer-reviewed. Also the analyses of the reported sequences can suggest false positives.

If the editor considers the paper for publication, I would advise for the author to tone down the claims that this is the best tip-dating done of the ancestor of SARS-CoV-2 (although it is the one with the larger dataset that I am aware) and discusses some of the issues that I think other researchers would have.

One point I find interesting is the identification of possible roots using the described methodology. However the author should describe the presence of the root in more exact terms and compare it with phylogenetically-estimated roots of the virus namely:

Forster et al. (2020) DOI: 10.1073/pnas.2004999117

Rito et al. (2020) DOI: doi.org/10.3390/microorganisms8111678

Gomez-Carballa et al. (2020) DOI: www.genome.org/cgi/doi/10.1101/gr.266221.120

And other relevant ones with different roots. I would find the comparison very interesting and I think other researchers would find it interesting (either matching any or not).

Author Response

I thank the reviewer for the insightful comments which bring several issues into focus and results in extensive rearrangement of the "Results and Discussion" section. My responses are in bold italic.

The manuscript “Dating the common ancestor from an NCBI tree of 83688 high-quality and full-length SARS-CoV-2 genomes” details the estimation the the ancestor of sampled SARS-CoV-2 using a large tree available in NCBI.

While the paper technically is sound, I worry about its novelty (tip-dating has been performed in other publications) and on an opposite view, I also worry about its major novelty: the early date obtained for the age of the ancestor of SARS-CoV-2.

Tip-dating with a known outgroup is straightforward. Tip-dating without a known outgroup is more complicated. 

The major advantage reported here is the fact that the author used a tree with data collected through 1.5 years. I wonder if this is an advantage. The circulating viruses 1.5 years after the first reported sequences is quite different from the early one in late 2019 and early 2020. As the author mentions “... some SARS-CoV-2 lineages appear to evolve faster or slower [4,19] either because of differential selection or mutation.” The history of the pandemic for 1.5 years shows a bias that could affect mutation rates estimates: often lineages that have a greater number of mutations are the ones that would have an increased spread with increased transmissibility against non-mutated lineages similar to the ancestors. I find more believable age estimates based on a small tree with only early sampled sequences (more similar to the SARS-CoV-2 ancestor lineage against a large tree with a great variability of lineages as reported here.

One cannot do tip-dating with only early sequences WITHOUT an outgroup. If one must work with early samples, then one need to use bat-derived or pangolin-derived sequences. In that case one is assuming little change of evolutionary rate over 50 or so years. For example, the three papers mentioned in the last comment by the reviewer all used bat/pangolin-derived viral sequences to root the tree.

I should also mention that, among programs that intended for dating without an outgroup, my programs TRAD and DAMBE are the only ones that can handle the large NCBI tree. TRAD is faster than DAMBE. One must use it (and other related programs) to be impressed.

The author even performed some analyses on this using only sampled genomes up to Jan. 15 2020, Mar. 21 2020, May 8 2020 and Apr. 3 2021 leading to consecutively earlier TMRCA with dates of  Dec. 13 2019, Dec. 4 2019, Oct. 20 2019 and Aug. 16 2019. This is an interesting trend. The author claims that “The larger the sample size, the more likely the sample will include early lineages, and the common ancestor will be dated earlier.” but that statement makes little sense as, from my knowledge, later trees do not earlier lineages, only descendants of the same early lineages detected up until early 2020.

Later lineages are not always descendants of early samples from Wuhan. This has become increasingly apparent since WHO's call for scientists to sequence archived samples collected around or before the viral outbreak in Wuhan (The call also includes animal surveillance and antibody test of blood samples two years back). I might illustrate this with a recently sequenced SARS-CoV-2 genome (GenBank accession MW795884). The viral genome was submitted on March 23, 2021 Utah Public Health Laboratories, based on a sample collected on Jan 13. 2020. Before this, people tend to think that the first Utah case was a Utah resident who came back to Utah from the cruise ship on Feb. 29, carrying the original D614 strain. In contract, this new genome with a collection time of Jan. 13, 2020 is a D614G strain, with all four characteristic mutations carried by other D614G strains. It differs from the reference strain in Wuhan by 25 mutations, suggesting that the D614G strain was already circulating in US when the Wuhan outbreak occurred. It also changed the perception that the D614G strain came to US from Europe. More likely it is the other way round. (Strangely, when I communicated this to Utah Public Health Laboratories and ask for more details about sample information at their web portal on Sept. 1, 2021, I received no reply. Instead, the GenBank record deleted the collection date on Sept. 4, 2021).

The author bases the reliability of the early age on a recent report from Italy claiming the presence of SARS-CoV-2 lineages since September 2019 in Northern Italy. I do not think that report has been properly peer-reviewed. Also the analyses of the reported sequences can suggest false positives.

My results are entirely independent from the two Italian studies, although I do expect more of such papers to be published in the future. These papers and my paper corroborate each other. By the way, I also mentioned in the manuscript that, while the first paper used the antibody approach which may have false positives, the second paper used the sequencing approach which is the gold standard for SARS-CoV-2 detection.

If the editor considers the paper for publication, I would advise for the author to tone down the claims that this is the best tip-dating done of the ancestor of SARS-CoV-2 (although it is the one with the larger dataset that I am aware) and discusses some of the issues that I think other researchers would have.

I did discuss the pitfalls of other dating studies that motivated my approach at the beginning of the paper, but added some elaboration later. 

One point I find interesting is the identification of possible roots using the described methodology. However the author should describe the presence of the root in more exact terms and compare it with phylogenetically-estimated roots of the virus namely:

Forster et al. (2020) DOI: 10.1073/pnas.2004999117

Rito et al. (2020) DOI: doi.org/10.3390/microorganisms8111678

Gomez-Carballa et al. (2020) DOI: www.genome.org/cgi/doi/10.1101/gr.266221.120

And other relevant ones with different roots. I would find the comparison very interesting and I think other researchers would find it interesting (either matching any or not).

The root is indicated by a red dot in Fig. 5. It is within the large Asian clade (those SARS-CoV-2 genomes sequenced beyond Asia can be traced to visitors to China or other Asian regions). This root position is unlikely to change given existing data, as long as new SARS-CoV-2 genomic sequences are mostly descendants of the existing viral lineages. However, the sequencing of archived samples may change the root position.

The root will be within the large Asian if the bat/pangolin-derived sequences remain the closest relatives to SARS-CoV-2. All these bat/pangolin-derived viral genomes share D614, so D614G most likely is derived, although the D614G change may have occurred much earlier than previously thought. For the same reason, if we use bat/pangolin-derived viral genomes to root the tree, then the root will also be within the large Asian clade. All three papers the reviewer listed above used bat/pangolin-derived viral genomes to root the tree, so the root was consistently found to be within the large Asian clade. There are several other papers taking the same approach. I have included these papers in discussion.

The problem of rooting with bat/pangolin-derived viral genomes is that the seemingly giant tree of SARS-CoV-2 essentially shrinks into a dot when the bat/pangolin-derived viral sequences are added. The reason is that SARS-CoV-2 sequences are, on the average, 99.9% identical, which is almost negligible compared to a sequence divergence of 96% between SARS-CoV-2 and the closest RaTG13 from bat. The consequence is that you can put the root in many different locations of the SARS-CoV-2 tree with the resulting tree lnL being essentially the same. Also, keep in mind that, in such an analysis, the evolutionary parameter estimates depend heavily on the branch leading from RaTG13 to SARS-CoV-2 because a lion share of substitutions occurred along this branch. Thus, one is extrapolating the evolutionary rate along this branch to the evolutionary rate among different SARS-CoV-2 lineages.

I discussed the weakness of this approach in the introduction of the paper, which motivated my approach of estimating the evolutionary rate and dating based on SARS-CoV-2 sequences only. My study corroborates these studies by showing that the root is within the large Asian clade without using bat/pangolin-derived viral genomes to root the tree, given existing genomic sequences in the NCBI tree.

I should mention that viral genomic divergence is typically gradual, so one should expect some viral genomes that are more closely related to SARS-CoV-2 than RaTG13, but are not descendants of existing SARS-CoV-2 lineages. If, after an extensive search, RaTG13 remains the closest relative of SARS-CoV-2, then the hypothesis of lab leak would gain steam.

Round 2

Reviewer 2 Report

Thanks for the replies to my raised concerns. I am happy with the responses and I think the manuscript should be published.